# LDMol: Text-to-Molecule Diffusion Model with Structurally Informative Latent Space

## Abstract

With the emergence of diffusion models as the frontline of generative models, many researchers have proposed molecule generation techniques with conditional diffusion models. However, the unavoidable discreteness of a molecule makes it difficult for a diffusion model to connect raw data with highly complex conditions like natural language. To address this, we present a novel latent diffusion model dubbed LDMol for text-conditioned molecule generation. LDMol comprises a molecule autoencoder that produces a learnable and structurally informative feature space, and a natural language-conditioned latent diffusion model. In particular, recognizing that multiple SMILES notations can represent the same molecule, we employ a contrastive learning strategy to extract feature space that is aware of the unique characteristics of the molecule structure. LDMol outperforms the existing baselines on the text-to-molecule generation benchmark, suggestion a potential that diffusion models can outperform autoregressive models in text data generation with a better choice of the latent domain. Furthermore, we show that LDMol can be applied to downstream tasks such as molecule-to-text retrieval and text-guided molecule editing, demonstrating its versatility as a diffusion model.

## 1 Introduction

Designing compounds with the desired characteristics is the essence of solving many chemical tasks. Inspired by the rapid development of generative models in the last decades, *de novo* molecule generation via deep learning models has been extensively studied. Diverse models have been proposed for generating molecules that agree with a given condition on various data modalities, including string representations (Segler et al., 2017), molecular graphs (Lim et al., 2020), and point clouds (Hoogeboom et al., 2022). The attributes controlled by these models evolved from simple chemical properties (Olivecrona et al., 2017; Gómez-Bombarelli et al., 2018) to complex biological activity (Staszak et al., 2022) and multi-objective conditioning (Li et al., 2018; Chang & Ye, 2024). More recently, as deep learning models' natural language comprehension ability has rapidly increased, there's a growing interest in molecule generation controlled by natural language (Edwards et al., 2022; Pei et al., 2023; Liu et al., 2024a; Su et al., 2022) which encompasses much broader and user-friendly controllable conditions.

Meanwhile, diffusion models (Song & Ermon, 2019; Ho et al., 2020) have emerged as a frontline of generative models over the past few years. Through a simple and stable training objective of predicting noise from noisy data (Ho et al., 2020), diffusion models have achieved highly realistic and controllable data generation (Dhariwal & Nichol, 2021; Karras et al., 2022; Ho & Salimans, 2021). Furthermore, leveraging that the score function of the data distribution is learned in their training (Song et al., 2021b), state-of-the-art image diffusion models enabled various applications on the image domain (Saharia et al., 2022; Kim & Ye, 2021; Chung et al., 2023). Inspired by the success of diffusion models, several papers suggested diffusion-based molecule generative models on various molecule domains including a molecular graph (Luo et al., 2023), strings like Simplified Molecular-Input Line-Entry System (SMILES) (Gong et al., 2024), and point clouds (Hoogeboom et al., 2022).

However, a discrepancy between molecule data and common data domains like images makes it hard to connect the diffusion models to molecule generation. Whereas diffusion models are deeply studied on a continuous data domain with Gaussian noise, any molecule modality has inevitable

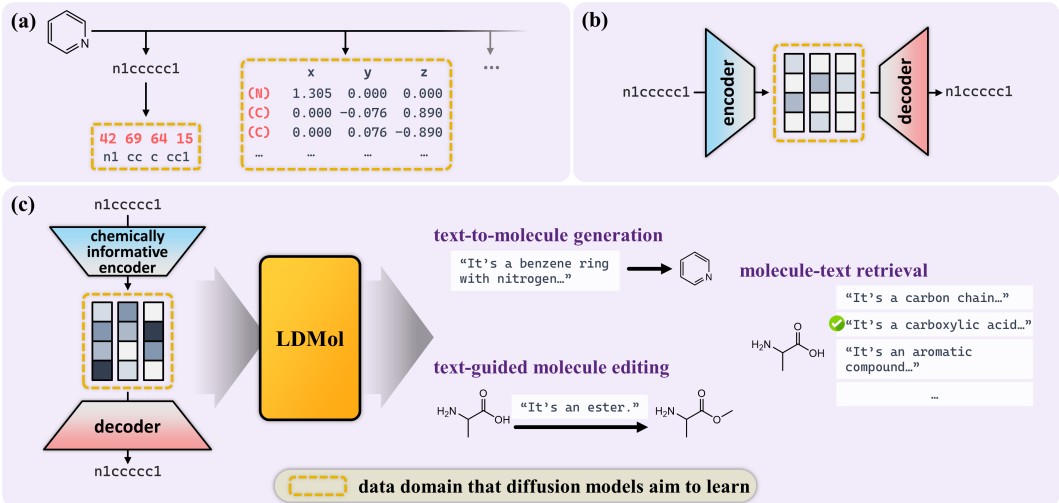

Figure 1: Different strategies of data domain selection for molecule diffusion models. (a) The model directly learns the raw representation of the molecule such as string format, point cloud coordinates, etc. (b) An autoencoder can be employed to let the generative model learn its latent distribution. (c) A regularized, chemically pre-trained encoder can provide a latent space learnable by external generative models.

discreteness such as atom and bond type, connectivity, and SMILES tokens as shown in Figure 1-(a). Naively converting discrete values into real numbers would ignore rich correlation within the components, and still retain certain restrictions on the data. As a result, diffusion models trained on raw molecule data often failed to faithfully follow the given conditions or showed poor data quality (*e.g.*, low validity) as the condition became more sophisticated like natural language. Most molecule diffusion models presented so far have used a few, relatively simple conditions to control, while major developments in text-conditioned molecule generative models were based on autoregressive models.

To overcome the limitation of the previous works (Xu et al., 2023) that mainly focused on resolving the discreteness of the molecular domain space by training with reconstruction loss (Figure 1-(b)), we aim to build a latent space encoder to extract rich and refined information about the molecule structure such as Figure 1-(c). Specifically, we design a novel Latent Diffusion Molecular generative model (LDMol) for text-conditioned molecule generation, trained on the latent space of the separately pre-trained molecule encoder. By preparing an encoder to provide a chemically useful and interpretable feature space, our model can more easily connect the molecule data with the highly complicated condition of natural texts. In the process, we suggest a novel contrastive learning strategy of training a molecular encoder to encode a structural invariant of the molecule.

The experimental results show that LDMol can outperform many state-of-the-art autoregressive models and generate valid SMILES that meet the input text condition. Considering SMILES as a variation of text data, we report one of the first diffusion models that successfully surpassed autoregressive models in text data generation. This suggests the possibility of improving existing diffusion models (Lovelace et al., 2024) for natural language through careful design of the latent space. Furthermore, LDMol can leverage the learned score function and be applied to several multi-modal downstream tasks such as molecule-to-text retrieval and text-guided molecule editing, without additional task-specific training. We summarize the contribution of this work as follows:

- We propose a latent diffusion model LDMol for text-conditioned molecule generation to generate valid molecules that are better aligned to the text condition. This approach demonstrates the potential of generative models for chemical entities in a latent space.

- We report the importance of preparing a chemically informative latent space for the molecule latent diffusion model, and suggest a novel contrastive learning method to train an encoder that captures the molecular structural characteristic.

- LDMol outperforms the text-to-molecule generation baselines, and its modeled conditional score function allows LDMol to retain the advanced attributes of diffusion models including various applications like molecule-to-text retrieval and text-guided molecule editing.

## 2 BACKGROUND

**Diffusion generative models.** Diffusion models first define a forward process that perturbs the original data, and generates the data from the known prior distribution by the learned reverse process of the pre-defined forward process. Ho et al. (2020) fixed their forward process by gradually adding Gaussian noise to the data, which can be formalized as follows:

$$q(x_t|x_{t-1}) = \mathcal{N}(x_t; \sqrt{1-\beta_t}x_{t-1}, \beta_t I) \tag{1}$$

where $\beta_t, t = 1, \ldots, T$ is a noise schedule. This definition of forward process allows us to sample $x_t$ directly from $q(x_t|x_0)$ as follows, where $\alpha_t = 1 - \beta_t$ and $\overline{\alpha}_t = \prod_{i=1}^{t} \alpha_i$:

$$x_t = \sqrt{\overline{\alpha}_t}x_0 + \sqrt{1-\overline{\alpha}_t}\epsilon, \text{ where } \epsilon \sim \mathcal{N}(0, I) \tag{2}$$

The model $\epsilon_\theta$ learns the reverse process $p(x_{t-1}|x_t)$ by approximating $q(x_{t-1}|x_t)$ with a Gaussian distribution $p_\theta(x_{t-1}|x_t) = \mathcal{N}(x_{t-1}; \mu_\theta(x_t, t), \sigma_t^2 I)$ where

$$\mu_\theta(x_t, t) = \frac{1}{\sqrt{\alpha_t}} \left( x_t - \frac{1-\alpha_t}{\sqrt{1-\overline{\alpha}_t}}\epsilon_\theta(x_t, t) \right) \tag{3}$$

which can be trained by minimizing the difference between $\epsilon$ and $\epsilon_\theta(x_t, t)$:

$$\theta^* = \arg\min_\theta \mathbb{E}_{x_0, t, \epsilon} ||\epsilon - \epsilon_\theta(x_t, t)||_2^2 \tag{4}$$

Once $\theta$ is trained, novel data can be generated with the learned reverse process $p_\theta$; starting from the random noise $x_T \sim \mathcal{N}(0, I)$, the output can be gradually denoised according to the modeled distribution of $p_\theta(x_{t-1}|x_t)$.

Various real-world data generation tasks require to generate data $x_0$ with a given condition $c$. To build diffusion models that can generate data from the conditional data distribution $q(x_0|c)$, the model that predicts the injected noise should also be conditioned by $c$.

$$\theta^* = \arg\min_\theta \mathbb{E}_{x_0, y, t, \epsilon} ||\epsilon - \epsilon_\theta(x_t, t, c)||_2^2 \tag{5}$$

Diffusion models can perform their diffusion process on the latent data space instead of the raw data when the raw data is too complex or high-dimensional (Vahdat et al., 2021; Rombach et al., 2022). Specifically, using a pre-trained pair of an encoder $\mathcal{E}(\cdot)$ and its corresponding decoder $\mathcal{D}(\cdot)$, datapoint $x$ is encoded into $z := \mathcal{E}(x)$ that is used to train the diffusion model. Generated latent $z'$ from the pre-trained diffusion model is transformed back to $x' := \mathcal{D}(z')$ on the original data domain. The latent diffusion model was also adopted for discrete texts (Lovelace et al., 2024), where the latent space was constructed by a denoising autoencoder.

**Conditional molecule generation.** As a promising tool for many important chemical and engineering tasks like *de novo* drug discovery and material design, conditional molecule generation has been extensively studied with various models including recurrent neural network (RNN)s (Segler et al., 2017), bidirectional RNN (Grisoni et al., 2020), graph neural networks (Lim et al., 2020), and variational autoencoders (Gómez-Bombarelli et al., 2018; Lim et al., 2018). With the advent of large and scalable pre-trained models with transformers (Vaswani et al., 2017), the controllable conditions became more abundant and complicated (Bagal et al., 2021; Chang & Ye, 2024). Recent works reached a text-guided molecule generation (Edwards et al., 2022; Su et al., 2022; Liu et al., 2024a) leveraging a deep comprehension ability for natural language, especially with recent emergence of large language model (LLM)s (Liu et al., 2024b).

Recent works attempted to import the success of the diffusion model into molecule generation. Several graph-based and point cloud-based works have built conditional diffusion models that could generate molecules with simple chemical and biological conditions (Hoogeboom et al., 2022; Luo et al., 2023; Trippe et al., 2023). Gong et al. (2024) attempted a text-conditioned molecule diffusion model trained on the sequence of tokenized SMILES indices. However, these models treated discrete molecules with continuous Gaussian diffusion, introducing arbitrary numeric values and suboptimal performances. Xu et al. (2023) employed an autoencoder to build a diffusion model on a smooth latent space, but its controllable conditions were still limited to several physiochemical properties.

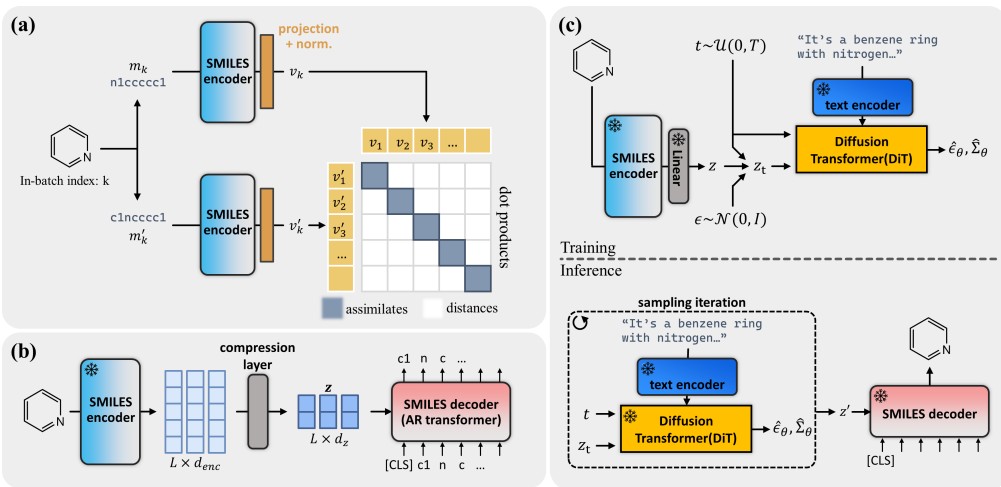

Figure 2: Overview of the proposed molecule autoencoder and the latent diffusion model. (a) SMILES encoder is trained with contrastive learning to construct latent space that embeds a structural characteristic. (b) After the SMILES encoder is prepared, a linear compression layer and an autoregressive decoder are trained to restore the encoder input. (c) The training and inference process of the latent diffusion model is conditioned by the output of the frozen external text encoder.

## 3 METHODS

In this section, we explain the overall model architecture and training procedure of the proposed LDMol, which are briefly illustrated in Figure 2. We first discuss the conditions the molecule encoder should meet and present our strategy to construct an encoder that can extract structural information from the raw molecule representation. Then we describe how we prepared a complete encoder-decoder pair with this encoder and trained a text-conditioned diffusion model on its latent space.

### 3.1 CHEMICALLY INFORMATIVE SMILES AUTOENCODER

The primary goal of introducing autoencoders for image latent diffusion models is to map raw images into a low-dimensional space which reduces the computation cost (Vahdat et al., 2021; Rombach et al., 2022). This is plausible because a high-resolution image has an enormous dimension on a pixel domain, yet each pixel contains little information.

In this work for molecule generation, we utilized SMILES, one of the most widely used molecule representations in chemical databases and benchmarks, and built a SMILES encoder to map raw SMILES strings into a latent vector. In this case, the role of our SMILES autoencoder has to be different from that of the autoencoders for images; a molecule structure can be fully expressed by only a sequence of $L$ integers for SMILES tokens, where $L$ is the maximum token length. However, each token carries significant information, and hidden interactions between these tokens are much more complicated than interactions between image pixels. Therefore, the SMILES encoder should focus more on extracting chemical meaning into the latent space, even if it results in a latent space with more dimension than the raw SMILES string.

A number of molecule encoders (Wang et al., 2019; Liu et al., 2024a; Zeng et al., 2022; Liu et al., 2023a) have been presented that can extract various useful chemical features including biochemical activity or human-annotated descriptions. Nonetheless, these molecule encoders aim to extract certain desired features rather than encode all the information about the molecule structure. Therefore the input cannot be fully restored from the model output.

Although autoencoders with appropriate regularization (*e.g.*, KL-divergence loss (Kingma, 2013; Gómez-Bombarelli et al., 2018)) provide a continuous and reconstructible molecular latent space, their encoder output is not guaranteed to possess the characteristic of the underlying molecular structure, beyond the minimal information to reconstruct the input string. To visualize this, we prepared a trained $\beta$-VAE (Higgins et al., 2017) and measured the feature distance between two SMILES from

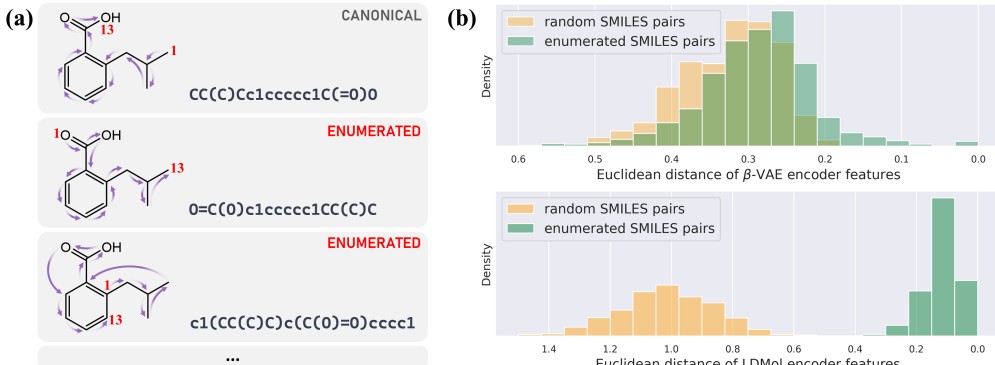

Figure 3: Behaviour of encoder features on SMILES enumeration. (a) Examples of SMILES enumeration with node traversal order. (b) Euclidean distance between features from $\beta$-VAE ($\beta = 0.001$) and our proposed encoder, with 1,000 random SMILES pairs and 1,000 enumerated SMILES pairs. The distance was rescaled by $1/\sqrt{d}$ where $d$ is a latent dimension size.

the same molecule obtained via SMILES enumeration (Bjerrum, 2017). Here, SMILES enumeration is the process of writing out all possible SMILES of the same molecule, as illustrated in Figure 3-(a). Figure 3-(b) shows that $\beta$-VAE had difficulties assimilating features from the same molecule compared to the one between random SMILES pairs, indicating that it couldn't capture the intrinsic features beneath the SMILES string. We believe this inconsistency comes from the discrepancy between reconstruction and comprehension of the data; even in the field of LLMs, it is known that even if the model has successfully learned the pattern of the natural language, the model doesn't understand its meaning and often hallucinates nonsensical output (Huang et al., 2023). Similarly, VAE could reconstruct the SMILES strings while not knowing what atoms and bonds the input implies. This inconsistency makes it difficult for later models that learn this latent space to figure out the connection between the latent and the molecule, which could eventually degrade the performance of the generative model with complex conditions. If a molecule encoder can truly provide a unique representation of the atoms and their connectivity, all SMILES representing the same molecule must be encoded into the same output and should be differentiated from the other random SMILES.

Accordingly, here we propose three conditions that our SMILES autoencoder's latent space has to satisfy: enable reconstruction of the input, have as small dimensions as possible, and embed chemical information that can be readily learned by diffusion models.

**Encoder design.** In this respect, we trained our SMILES encoder with contrastive learning (Figure 2-(a)), which aims to learn better representation by assimilating features containing similar information (*i.e.* positive pair) and distancing semantically unrelated features (*i.e.* negative pair). We defined two enumerated SMILES from the same molecule as a positive pair and two SMILES from different molecules as a negative pair.

Here, we argue that the proposed contrastive learning with SMILES enumeration can train the encoder to encapsulate the structural information of the input molecule. Contrastive learning learns invariant for the augmentations applied on positive pairs (Zhang & Ma, 2022), and it is known that a good augmentation should reduce as much mutual information between positive pairs as possible while preserving relevant information (Tian et al., 2020). Meanwhile, enumerated SMILES of the same molecule are obtained by traversing the nodes and edges in the molecular graph with a different visiting order. Therefore, to detect all possible enumerated SMILES and find SMILES-enumeration-invariant, the model has to understand the entire connectivity between atoms. This makes the encoder output a unique characteristic that captures the overall molecule structure. Compared to the hand-crafted augmentations previously presented for molecule contrastive learning (You et al., 2020), enumerated SMILES pairs have less mutual information since we utilize all possible variations in the SMILES format. And since all enumerated SMILES are guaranteed to represent an identical molecule, there is no relevant information loss during the augmentation. Figure 3-(a) confirms that our LDMol trained with the contrastive learning with SMILES enumeration now correctly assimilates features from the same molecule compared to the one between random SMILES pairs.

Specifically, a SMILES string is fed into the SMILES encoder $\mathcal{E}(\cdot)$ with a special "[SOS]" token which denotes the start of the sequence. For a batch of $N$ input SMILES $M = \{m_1, m_2, \ldots, m_k, \ldots, m_N\}$, we prepare a positive pair SMILES $m'_k$ for each $m_k$ by SMILES enumeration to construct $M' = \{m'_1, m'_2, \ldots, m'_k, \ldots, m'_N\}$. After $M$ and $M'$ are passed through the SMILES encoder, we fed each SMILES' output vector corresponding to the [SOS] token into an additional linear projection and normalization layer, denoting its output as $v_k$ and $v'_k (k = 1, 2, \ldots, N)$. Assimilating $v_k$ and $v'_k$ from the positive pairs and distancing the others can be done by minimizing the following InfoNCE loss (Chen et al., 2020).

$$\mathcal{L}_{contrastive}(M, M') = -\sum_{k=1}^{N} \log \frac{\exp(v_k \cdot v'_k / \tau)}{\sum_{i=1}^{N} \exp(v_k \cdot v'_i / \tau)} \tag{6}$$

Here, $\tau$ is a positive temperature parameter. To utilize a symmetric loss against the input, we trained our encoder with the following loss function.

$$\mathcal{L}_{enc}(M, M') = \mathcal{L}_{contrastive}(M, M') + \mathcal{L}_{contrastive}(M', M) \tag{7}$$

While our encoder was pre-trained to map structurally similar molecules to similar vectors, we found that the encoder tried to map stereoisomers to almost the same point in the latent space; indeed, in terms of structural differences, stereoisomers are the most challenging molecules to distinguish. To resolve this, if the training data has a stereoisomer, we consider it as "hard-negative" samples and include it in the loss calculation batch with the original data. As such, we can enable the ability to differentiate between different stereoisomers.

**Compressing the latent space.** The pre-trained SMILES encoder maps a molecule into a vector of size $[L \times d_{enc}]$, where $d_{enc}$ is a feature size of the encoder. To avoid the curse of dimensionality and construct a more learnable feature space for diffusion models, we additionally employed a linear compression layer $f(\cdot)$ (Figure 2-(b)) to reduce the dimension from $[L \times d_{enc}]$ to $[L \times d_z]$. The range of this linear layer output is a target domain of our latent diffusion model.

**Decoder design.** When a SMILES $m$ is passed through the SMILES decoder and the compression layer, the SMILES decoder reconstructs $m$ from $f(\mathcal{E}(m))$. Following many major works that treated SMILES as a variant of language data (Segler et al., 2017; Chang & Ye, 2024; Ross et al., 2022), we built an autoregressive transformer (Vaswani et al., 2017) as our decoder (Figure 2-(b)) which is widely used to successfully generate sequential data (Brown et al., 2020). Specifically, starting from the [SOS] token, the decoder predicts the next SMILES token using information from $f(\mathcal{E}(m))$ with cross-attention layers. When $\{t_0, t_1, \ldots, t_n\}$ is the token sequence of $m$, the decoder is trained to minimize the next-token prediction loss described as Eq. (8). Here, the decoder and the compression layer are jointly trained while the encoder's parameter is frozen.

$$\mathcal{L}_{dec} = -\sum_{i=1}^{n} \log p(t_n | t_{0:n-1}, f(\mathcal{E}(m))) \tag{8}$$

As long as the encoder output contains complete information of the molecule structure, the decoder can reconstruct the frozen encoder input. After being fully trained, the decoder was able to reconstruct roughly 98% of the SMILES encoder input.

## 3.2 TEXT-CONDITIONED LATENT DIFFUSION MODEL

As shown in Figure 2-(c), our diffusion model learns the conditional distribution of the SMILES latent $z$ whose dimension is $[L \times d_z]$. In the training phase, a molecule in the training data is mapped to the latent $z$ and applied a forward noising process into $z_t$ with randomly sampled diffusion timestep $t$ and injected noise $\epsilon$. A diffusion model predicts the injected noise from $z_t$, conditioned by the paired text description via a frozen external text encoder. In the inference phase, the diffusion model iteratively generates a new latent sample $z'$ from a given text condition, which is then decoded to a molecule via the SMILES decoder.

Since most contributions to diffusion models were made in the image domain, most off-the-shelf diffusion models have the architecture of convolution-based Unet (Ronneberger et al., 2015). However, introducing the spatial inductive bias of Unet cannot be justified for the latent space of our encoder. Therefore we employed DiT (Peebles & Xie, 2023) architecture, one of the most successful approaches to transformer-based diffusion models for more general data domain. Specifically, we

utilized a $\text{DiT}_{base}$ model with minimal modifications to handle text conditions with cross-attention, where more details can be found in Section A.1.

Text-based molecule generation requires a text encoder to process natural language conditions. Existing text-based molecule generation models trained their text encoder from scratch (Pei et al., 2023), or utilize a separate encoder model pre-trained on scientific domain corpora (Beltagy et al., 2019). In this work, we took the encoder part of $\text{MolT5}_{large}$ (Edwards et al., 2022) as our text encoder.

### 3.3 IMPLEMENTATION DETAILS

The pre-training of the SMILES encoder and the corresponding decoder was done with 10,000,000 general molecules from PubChem (Kim et al., 2023). To ensure enough batch size for negative samples (He et al., 2020), we build a memory queue that stores $Q$ recent input and use them for the encoder training. The SMILES tokenizer vocabulary consists of 300 tokens, which were obtained from the pre-training data SMILES corpus using the BPE algorithm (Gage, 1994). We only used SMILES data that does not exceed a fixed maximum token length $L$.

To train the text-conditioned latent diffusion model, we gathered three existing datasets of text-molecule pair: PubchemSTM curated by Liu et al. (2023a), ChEBI-20 (Edwards et al., 2021), and PCdes (Zeng et al., 2022). Only a train split for each dataset was used for the training, and pairs that appear in the test set for the experiments are additionally removed. We also used 10,000 molecules from ZINC15 (Sterling & Irwin, 2015) without any text descriptions, which helps the model learn the common distribution of molecules. When these unlabeled data were fed into the training model, we used a pre-defined null text which represents the absence of text condition.

The latent diffusion model was trained with the training loss suggested by Dhariwal & Nichol (2021). To take advantage of classifier-free guidance (Ho & Salimans, 2021), we randomly replaced 3% of the given text condition with the null text during the training. The sampling iteration in the inference stage used DDIM-based (Song et al., 2021a) 100 sampling steps with a classifier-free guidance. More detailed training hyperparameters can be found in Appendix A.2.

## 4 EXPERIMENTS

### 4.1 TEXT-CONDITIONED MOLECULE GENERATION

In this section, we evaluated the trained LDMol's ability to generate molecules that agree with the given natural language conditions. First, we generated molecules with LDMol using the text captions in the ChEBI-20 test set and compared them with the ground truth. The metrics we've used are as follows: SMILES validity, BLEU score (Papineni et al., 2002) and Levenshtein distance between two SMILES, Tanimoto similarity (Bajusz et al., 2015) between two SMILES with three different fingerprints (MACCS, RDK, Morgan), the exact match ratio, and Frechet ChemNet Distance (FCD) (Preuer et al., 2018).

We note that since our diffusion model sampling starts from randomly sampled noise from a prior distribution, the output from deterministic DDIM sampling still has stochasticity. However, with a highly detailed text prompt that can specify a single correct answer, we discovered that LDMol was powerful enough to outperform many autoregressive models with randomly sampled initial points. Also, the performance difference was marginal between starting from random noise and starting from a fixed point (*e.g.*, zero vector). We tested different scales for the classifier-free guidance scale $\omega$ in the sampling process and found $\omega = 2.5$ works best (See Section B.1).

Table 1 contains the performance of LDMol and other baselines for text-to-molecule generation on the ChEBI-20 test set. Including both transformer-based autoregressive models and diffusion-based models, LDMol outperformed the existing models in almost every metric. While few models showed higher validity than ours, they showed a lower agreement between the output and the ground truth, which we insist is a more important role of generative text-to-molecule models. One thing to emphasize is that $\text{MolT5}_{large}$ uses the same text encoder with LDMol, yet there's a significant performance difference between the two models. We believe this is because our constructed latent

Table 1: Benchmark results of text-to-molecule generation on ChEBI-20 test set. The best performance for each metric was written in **bold.** The "Family" column denotes whether the model is AR(autoregressive model) or DM(diffusion model).

| Model | Family | Validity↑ | BLEU↑ | Levenshtein↓ | MACCS FTS↑ | RDK FTS↑ | Morgan FTS↑ | Match↑ | FCD↓ |
|---|---|---|---|---|---|---|---|---|---|
| Transformer | AR | 0.906 | 0.499 | 57.660 | 0.480 | 0.320 | 0.217 | 0.000 | 11.32 |
| GIT-Mol (Liu et al., 2024a) | AR | 0.928 | 0.756 | 26.315 | 0.738 | 0.582 | 0.519 | 0.051 | - |
| T5$_{base}$ (Raffel et al., 2020) | AR | 0.660 | 0.765 | 24.950 | 0.731 | 0.605 | 0.545 | 0.069 | 2.48 |
| MolT5$_{base}$ (Edwards et al., 2022) | AR | 0.772 | 0.769 | 24.458 | 0.721 | 0.588 | 0.529 | 0.081 | 2.18 |
| T5$_{large}$ | AR | 0.902 | 0.854 | 16.721 | 0.823 | 0.731 | 0.670 | 0.279 | 1.22 |
| MolT5$_{large}$ | AR | 0.905 | 0.854 | 16.071 | 0.834 | 0.746 | 0.684 | 0.311 | 1.20 |
| MolXPT (Liu et al., 2023b) | AR | 0.983 | - | - | 0.859 | 0.757 | 0.667 | 0.215 | 0.45 |
| bioT5 (Pei et al., 2023) | AR | **1.000** | 0.867 | 15.097 | 0.886 | 0.801 | 0.734 | 0.413 | 0.43 |
| bioT5+ (Pei et al., 2024) | AR | **1.000** | 0.872 | 12.776 | 0.907 | 0.835 | 0.779 | 0.522 | 0.35 |
| TGM-DLM (Gong et al., 2024) | DM | 0.871 | 0.826 | 17.003 | 0.854 | 0.739 | 0.688 | 0.242 | 0.77 |
| LDMol | DM | 0.941 | **0.926** | **6.750** | **0.973** | **0.950** | **0.931** | **0.530** | **0.20** |

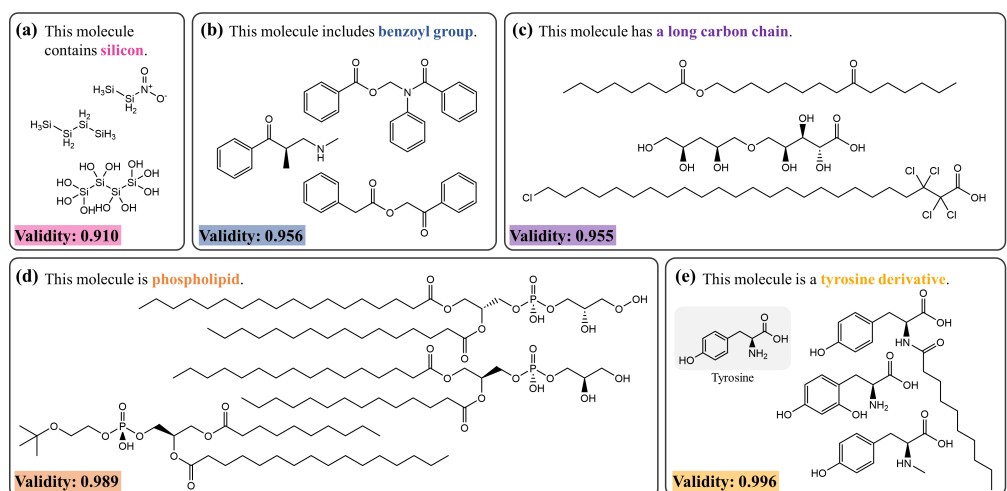

Figure 4: Examples of the generated molecules by LDMol with various text conditions, with validity on 1,000 generated samples.

space is much easier to align with the same textual information, compared to the token sequence used by transformer-based models.

To demonstrate the LDMol's molecule generalization ability with more broad and general text inputs, we analyzed the generated output with several hand-written prompts. These input prompts were not contained in the training data and were relatively vague and high-level so that many different molecules could satisfy the condition. Figure 4 shows the samples of generated molecules from LDMol with several input prompt examples. We found that LMDol can generate molecules with high validity that follow the various levels of input conditions for specific atoms(a), functional groups(b), molecular substructure(c), compound class(d), and substance names(e). The validity was calculated by the number of valid SMILES over 1,000 generated samples, above 0.9 for most scenarios we tested. Considering that these short, broad, and hand-written text conditions are distinct from the text conditions in the training dataset, we've concluded that our model is able to learn the general relation between natural language and molecules.

## 4.2 APPLICATIONS TOWARD DOWNSTREAM TASKS

Well-trained diffusion models learned the score function of a data distribution, which enables high applicability to various downstream tasks. The state-of-the-art image diffusion models have shown their versatility in image editing (Meng et al., 2022; Hertz et al., 2023), classification (Li et al., 2023), retrieval (Jin et al., 2023), inverse problems like inpainting and deblurring (Chung et al., 2023), image personalization (Ruiz et al., 2023), etc. To demonstrate LMDol's potential versatility as a diffusion model, we applied the pre-trained LDMol to the molecule-to-text retrieval and text-guided molecule editing task.

Table 2: 64-way accuracy in % on molecule-to-text retrieval task. For LDMol, $n$ is a number of iterations where $||\hat{\epsilon}_\theta - \epsilon||_2^2$ was calculated. The best performance for each task is written in **bold.**

| Model | PCdes test set | | MoMu test set | |
|---|---|---|---|---|
| | sentence-level | paragraph-level | sentence-level | paragraph-level |
| SciBERT (Beltagy et al., 2019) | 50.4 | 82.6 | 1.38 | 1.38 |
| KV-PLM (Zeng et al., 2022) | 55.9 | 77.9 | 1.37 | 1.51 |
| MoMu-S (Su et al., 2022) | 58.6 | 80.6 | 39.5 | 45.7 |
| MoMu-K | 58.7 | 81.1 | 39.1 | 46.2 |
| MoleculeSTM (Liu et al., 2023a) | - | 81.4 | - | 67.6 |
| MolCA (Liu et al., 2024c) | - | 86.4 | - | 73.4 |
| LDMol($n$=10) | 60.7 | 90.2 | 66.4 | 84.8 |
| LDMol($n$=25) | **62.2** | **90.3** | **78.4** | **87.1** |

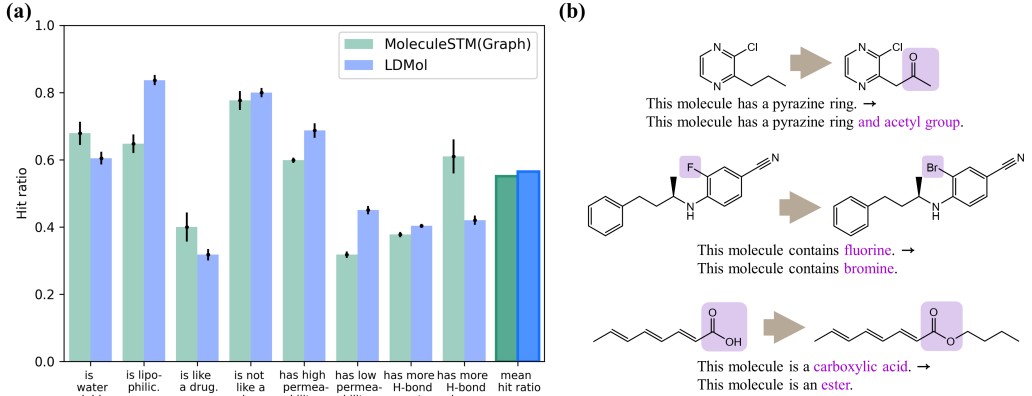

Figure 5: (a) Hit ratio of molecule editing by LDMol and MoleculeSTM (Liu et al., 2023a) in eight scenarios. Following MoleculeSTM, each scenario was applied to 200 randomly sampled molecules from ZINC15, and the mean and standard variation on three separate runs were plotted. In the figure, we omitted "*This molecule*" in front of the actual prompts, and abbreviated "*hydrogen bonding*" to "*H-bond*". (b) Examples of text-guided molecule editing with LDMol. The difference between the source text and the target text, and the corresponding region is colored in purple.

**Molecule-to-text retrieval.** Our approach to molecule-to-text retrieval is similar to the idea of using a pre-trained diffusion model as a classifier (Li et al., 2023): LDMol takes each candidate text with a query molecule's noised latent, and retrieves the text that minimizes the noise estimation error $||\hat{\epsilon}_\theta - \epsilon||_2^2$ between the injected noise $\epsilon$ and the predicted noise $\hat{\epsilon}_\theta$ (see Appendix A.3 for more details). Since this process has randomness due to the stochasticity of $t$ and $\epsilon$, we repeated the same process $n$ times with resampled $t$ and $\epsilon$ and used a mean error to minimize the performance variance.

We measured a 64-way in-batch retrieval accuracy of LDMol using two different test sets: PCdes test split and MoMu retrieval dataset curated by Su et al. (2022), where the result with other baseline models are listed in Table 2. Only one randomly selected sentence in each candidate description was used for the retrieval in the "sentence-level" column, and all descriptions were used for the "paragraph-level" column. LDMol achieved a higher performance in all four scenarios compared to the previously presented models and maintained its performance on a relatively out-of-distribution MoMu test set with minimal accuracy drop. LDMol became more accurate as the number of function evaluations increased, and the improvement was more significant at the sentence-level retrieval and out-of-distribution dataset. The actual examples from the retrieval result can be found in Appendix B.2.

**Text-guided molecule editing.** We applied a method of Delta Denoising Score (DDS) (Hertz et al., 2023), which was originally suggested for text-guided image editing, to see whether LDMol can be used to optimize a source molecule to match a given text. Using two text prompts that describe the source data $z_{src}$ and the desired target, DDS presents how a text-conditioned diffusion model can modify $z_{src}$ into a new data $z_{tgt}$ that follows the target text prompt (more detailed procedures can be found in Appendix A.3).

Table 3: Quantitative results of the ablation studies.

| models | Autoencoder | Text-to-molecule generation | | |
|---|---|---|---|---|
| | Recon. Acc.↑ | Validity↑ | Match↑ | FCD↓ |
| LDMol w/o contrastive learning(*i.e.* naive AE) | 1.000 | 0.019 | 0.000 | 58.60 |
| LDMol w/o compression layer | 0.964 | 0.022 | 0.000 | 67.93 |
| LDMol w/o stereoisomer hard-negative | 0.891 | 0.939 | 0.278 | 0.24 |
| naive AE + KL regularization($\beta = 0.001$) | 0.999 | 0.847 | 0.492 | 0.34 |
| LDMol | 0.983 | 0.941 | 0.530 | 0.20 |

We imported a method of DDS on LDMol's molecule latent to edit a given molecule to match the target text, with several prepared editing prompts that require the model to change certain atoms, substructures, and intrinsic properties from the source molecule. Figure 5-(a) shows that LDMol had comparable performance with a previously suggested text-guided molecule editing model (Liu et al., 2023a), with a higher hit ratio in five out of eight scenarios. Actual case studies with editing sources and results are listed in Figure 5-(b), where the editing results successfully modified the input molecule towards the target prompt with minimal corruption of the unrelated region.

## 4.3 ABLATION STUDIES

We've conducted ablation studies for each design choice of the proposed LDMol to analyze and emphasize their role. Each model is pre-trained with the same number of DiT training iterations. The results of the ablation studies are listed in Table 3.

When we remove the contrastive encoder pre-training objective and construct the molecule latent space with a naive autoencoder, the diffusion model completely fails to learn the latent distribution to generate valid SMILES. A similar problem occurred when we didn't introduce a compression layer since the dimension of the latent space was too big for the diffusion model to learn. When stereoisomers were not utilized as hard negative samples in the contrastive encoder training, the constructed latent space was not detailed enough to specify the input, which degraded the reconstruction accuracy of the autoencoder.

An interesting case is the comparison against the autoencoder with KL-divergence loss: it has reconstructible latent space and showed a text-to-molecule generation match ratio of 0.492, which outperforms several baseline models in Table 1. This performance shows the ability of diffusion models on the continuous data domain which can be comparable to the autoregressive models. However, its overall metric is still lower than the proposed LDMol, with especially low validity and FCD. We insist that this gap comes from the structurally informative latent space of LDMol which is easier for the diffusion model to learn the correlation between the latent space and the condition.

## 5 CONCLUSION

In this work, we presented a text-conditioned molecule diffusion model LDMol that runs on a chemical latent space reflecting structural information. By introducing the deeply studied paradigm of the latent diffusion model into the molecule domain with minimal modifications, LDMol retains many advanced attributes of diffusion models that enable various applications.

Despite the noticeable performances of LDMol, it still has limitations that can be improved: Since the text captions of our obtained paired data mostly describe structural features, LDMol still struggles to follow some text conditions such as complex biological properties. Nonetheless, we expect that the LDMol's performance could be improved further with the emergence of richer text-molecule pair data and more powerful text encoders. Moreover, combining physiochemical and biological annotations on top of the structurally informative latent space is a promising future work that can ease the connection between molecules and text conditions.

We believe that our approach could inspire tackling various chemical generation tasks using latent space, not only text-conditioned but also many more desired properties such as biochemical activity. Especially, we expect LDMol to be a starting point to bridge achievements in the state-of-the-art diffusion model into the chemical domain.

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

## A EXPERIMENTAL DETAILS

### A.1 DiT BLOCK FOR SMILES LATENT DIFFUSION MODEL WITH TEXT CONDITIONS

The DiT block architecture of the class-conditioned image diffusion model published by Peebles & Xie (2023) is shown in Figure 6-(a). The noised input image latent $z_t$ is passed through a patch embedding layer and spatially flattened to be fed into the DiT block. The condition embedding $y$ and diffusion timestep embedding $t$ are incorporated into the model prediction via adaptive layer norm. The dimension of $t$ and $y$ are both $[B \times F]$, where $B$ is the batch size and $F$ is the number of features.

In the case of LDMol, the input latent $z_t$ with dimension $[B \times L \times F]$ is already spatially one-dimensional, we simply pass it through a linear layer to prepare DiT block input. Also, the text condition feature we've used has a much higher dimension of $[B \times L' \times F]$ where $L'$ is the token length of the text condition. Therefore, we stacked a cross-attention layer for text condition features after each self-attention layer, as shown in Figure 6-(b).

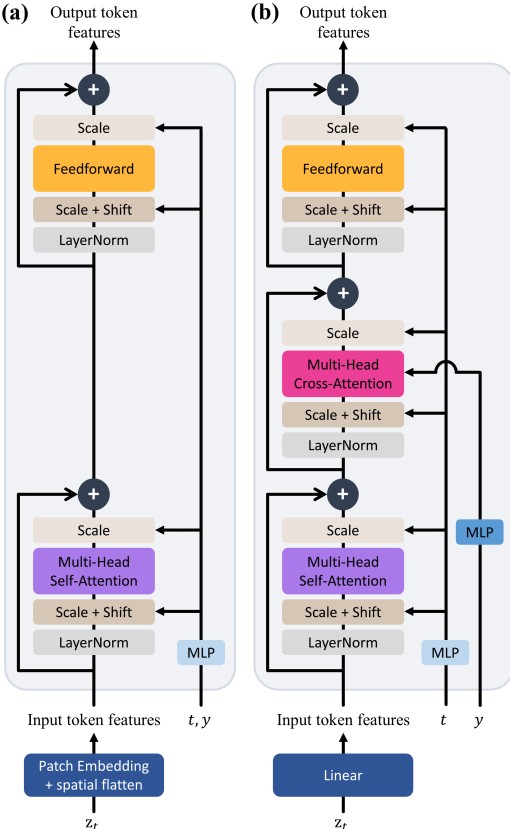

Figure 6: Input embedding layer and DiT blocks in the (a) originally published DiT and (b) LDMol.

### A.2 MODEL HYPERPARAMETERS AND TRAINING SETUP

Detailed hyperparameters on the model architecture are listed in Table 4, with settings on the model training procedure. Here, we provide the results of several ablation studies to support our selection of the used hyperparameters.

It is known that lower temperature $\tau$ in contrastive learning penalizes hard negatives more strongly, making the learned feature more sensitive to fine-grain details (Wang & Liu, 2021). We considered this as a desirable property for our latent space and used a small tau of 0.07. When we used too big $\tau$ of 0.15 as shown in Figure 7, it reduced the model's ability to distinguish different molecules and made the training loss converge to a much higher value.

Table 4: The choice of the model hyperparameters and training setup.

| hyperparameters | |
| --- | --- |
| $L$ | 128 |
| $d_{enc}$ | 1024 |
| $d_z$ | 64 |
| $\tau$ | 0.07 |
| $Q$ | 16384 |
| training setup | |
| optimizer | autoencoder: AdamW, DiT: Adam |
| learning rate | autoencoder: cosine annealing(1e-4→1e-5), DiT: 1e-4 |
| batch size per GPU | encoder: 64, decoder: 128, DiT: 64 |

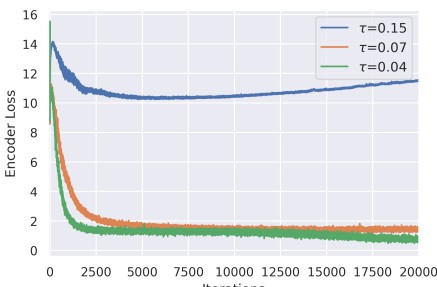

| Methods | | Recon. Acc. |
| --- | --- | --- |
| Contrastive learning on compressed latent with $d_z$=64 | | 0.084 |
| Contrastive learning on $d_{enc}$=1024 | compression with $d_z$=32 | 0.948 |
| | compression with $d_z$=64 | 0.980 |
| | compression with $d_z$=128 | 0.989 |

Table 5: SMILES reconstruction accuracy of various trained autoencoders, with 1,000 unseen SMILES.

Figure 7: The encoder loss convergence with different temperature parameter $\tau$.

Table 5 lists the LDMol autoencoder's SMILES reconstruction accuracy in various autoencoder training strategies. When we apply contrastive loss directly into the compressed latent domain, the encoder fails to capture informative features, makes the decoder couldn't reconstruct the input molecule. In the scenario of adding linear compression after contrastive training with $d_{enc}$=1024, we observed an error rate of more than 5% for the compression size of $d_z$=32. Compression with $d_z$=128 slightly increased the reconstruction accuracy compared to $d_z = 64$, but the training time for the subsequent diffusion model rapidly increased. Considering that the failed 2% for the current model were mostly very long molecules, we concluded that $d_z = 64$ is sufficient for our model.

A.3 MORE DETAILED PROCEDURE FOR LDMOL'S APPLICATION ON DOWNSTREAM TASKS

**Algorithm 1** Molecule-to-Text Retrieval with LDMol

**Require:** $z, \mathcal{C} = \{c_i\}_{i=1}^B, n \in \mathbb{N}^+$
1: Initialize $\texttt{Errors}[c_i] = 0$ **for** $i = 1$ **to** $B$
2: **for** $\texttt{iter} = 1$ **to** $n$ **do**
3:     $t \sim U[0, T], \epsilon \sim \mathcal{N}(0, I)$
4:     $z_t = \sqrt{\overline{\alpha}_t}z + \sqrt{1 - \overline{\alpha}_t}\epsilon$
5:     **for** $i = 1$ **to** $B$ **do**
6:         $\texttt{Errors}[c_i] \mathrel{+}= ||\hat{\epsilon}_\theta(z_t, t, c_i) - \epsilon||_2^2$
7:     **end for**
8: **end for**
9: **Return** $\operatorname{argmin}_{c_i \in \mathcal{C}} \texttt{Errors}[c_i]$

**Algorithm 2** Text-guided Molecule Editing with LDMol

**Require:** $z_{src}, c_{src}, c_{tgt}, N \in \mathbb{N}^+, \gamma > 0, \omega \geq 1, \mathcal{D}$
1: Initialize $z_{tgt} = z_{src}$
2: **for** $\texttt{iter} = 1$ **to** $N$ **do**
3:     $t \sim U[0, T], \epsilon \sim \mathcal{N}(0, I)$
4:     $z_{t,src} = \sqrt{\overline{\alpha}_t}z_{src} + \sqrt{1 - \overline{\alpha}_t}\epsilon$
5:     $z_{t,tgt} = \sqrt{\overline{\alpha}_t}z_{tgt} + \sqrt{1 - \overline{\alpha}_t}\epsilon$
6:     $\epsilon_{\theta,src}^\omega = (1 - \omega)\epsilon_\theta(z_{t,src}, t, \varnothing) + \omega\epsilon_\theta(z_{t,src}, t, c_{src})$
7:     $\epsilon_{\theta,tgt}^\omega = (1 - \omega)\epsilon_\theta(z_{t,tgt}, t, \varnothing) + \omega\epsilon_\theta(z_{t,tgt}, t, c_{tgt})$
8:     $z_{tgt} = z_{tgt} - \gamma(\epsilon_{\theta,tgt}^\omega - \epsilon_{\theta,src}^\omega)$
9: **end for**
10: **Return** $\mathcal{D}(z_{tgt})$

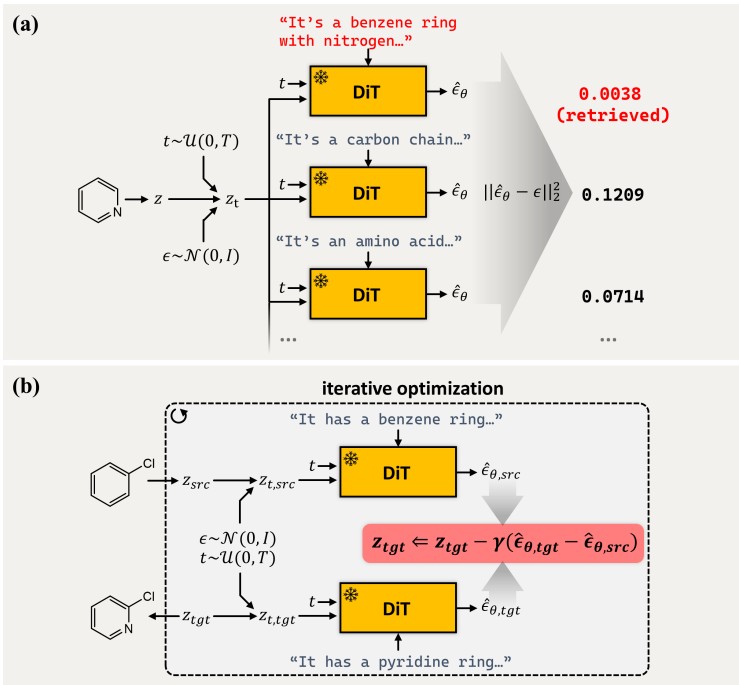

Figure 8: Overall pipeline for the downstream task applications of LDMol. (a) Molecule-to-text retrieval. (b) Text-guided molecule editing. The SMILES autoencoder and the text encoder are not drawn in this figure.

Figure 8-(a) and Algorithm 1 show the LDMol's molecule-to-text retrieval process with a given query molecule and text candidates $\mathcal{C} = \{c_i\}_{i=1}^{B}$. A given query molecule is converted to a latent $z$, and then a forward noise process is applied with a randomly sampled timestep $t$ and noise $\epsilon$. This $z_t$ is fed to LDMol with each candidate $c_i$, and the candidate that minimizes the loss $||\hat{\epsilon}_\theta(z_t, t, c_i) - \epsilon||_2^2$ between $\epsilon$ and the output noise $\hat{\epsilon}_\theta(z_t, t, c_i)$ is retrieved. To minimize the variance from the stochasticity of $t$ and $\epsilon$, the same process can be repeated $n$ times with resampled $t$ and $\epsilon$ to use a mean loss.

Figure 8-(b) and Algorithm 2 illustrate the DDS-based molecule editing with LDMol. Specifically, DDS requires source data $z_{src}$, target data $z_{tgt}$ which is initialized to $z_{src}$, and their corresponding source and target text descriptions $\{c_{src}, c_{tgt}\}$. We apply the forward noise process to $z_{src}$ and $z_{tgt}$ using the same randomly sampled $t$ and $\epsilon$ to get $z_{t,src}$ and $z_{t,tgt}$. These are fed into the pre-trained LDMol with their corresponding text, where we denote the output noise as $\hat{\epsilon}_{\theta,src}$ and $\hat{\epsilon}_{\theta,tgt}$. Finally, $z_{tgt}$ is modified towards the target text by optimizing it to the direction of $(\hat{\epsilon}_{\theta,tgt} - \hat{\epsilon}_{\theta,src})$ with a learning rate $\gamma$. Here, $\hat{\epsilon}_{\theta,tgt}$ and $\hat{\epsilon}_{\theta,tgt}$ can be replaced with the classifier-free-guidanced noises, utilizing the output with the null text and the guidance scale $\omega$. $z_{tgt}$ is decoded back as the editing output after the optimization step is iterated $N$ times. In Figure 5-(a), where we applied the same scenario to a batch of molecules, we used null text as $c_{src}$ since it's impractical to prepare a source prompt for each molecule. The hyperparameters $\{N, \gamma, \omega\}$ are fixed for each scenario, where every choice is in the range of $100 \leq N \leq 200$, $0.1 \leq \gamma \leq 0.3$ and $2.0 \leq \omega \leq 4.5$. For the illustrated examples in Figure 5-(b), we repeated DDS iterations with $N = 150$, $\gamma = 0.1$ and $\omega = 2.5$.

## B  ADDITIONAL RESULTS

### B.1  EFFECT OF CLASSIFIER-FREE GUIDANCE SCALE ON LDMOL SAMPLE QUALITY

Figure 9 plots the LDMol's text-to-molecule generation performance on the ChEBI-20 test set, with different classifier-free guidance scale $\omega$ in the sampling process. Starting from $\omega = 1.0$ which is equivalent to a naive conditional generation, we observed the overall sample quality is improved

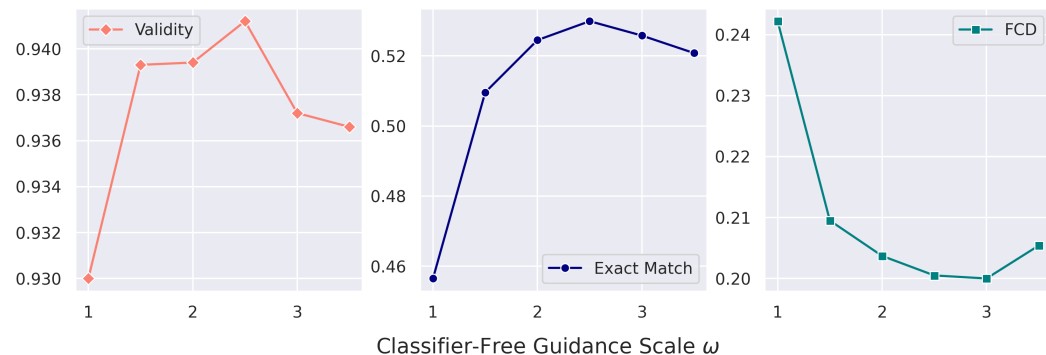

Figure 9: Text-to-molecule generation performance of LDMol against different classifier-free guidance scales.

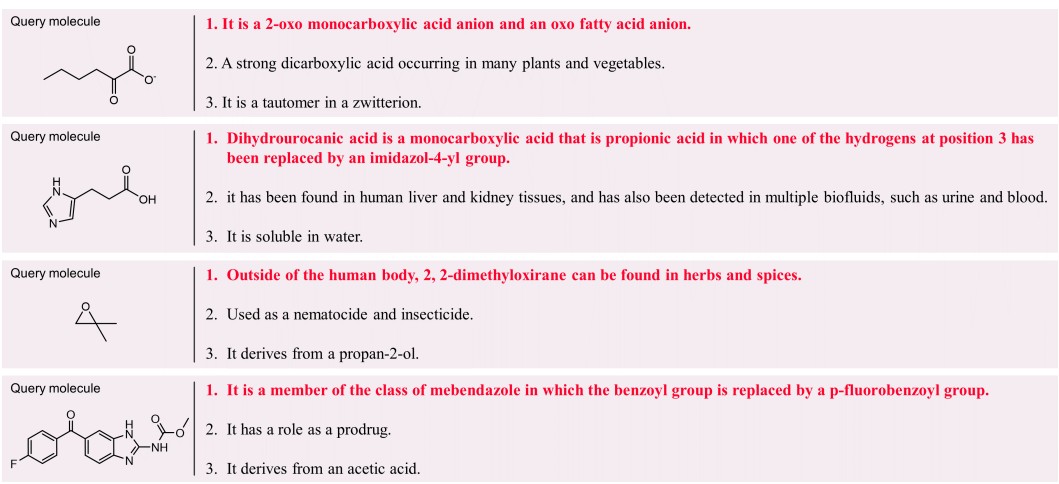

Figure 10: The examples of molecule-to-text retrieval result on the PCdes test set. Three sentences with the lowest noise estimation error were retrieved for each query molecule.

as $\omega$ increases but collapses for too big $\omega$. This agrees with the well-known observation on image diffusion models, and we decided to use $\omega = 2.5$ for the text-to-molecule generation with LDMol.

## B.2 MOLECULE-TO-TEXT RETRIEVAL EXAMPLES

Figure 10 contains examples of molecule-to-text retrieval results with molecules from the PCdes test set. The retrieval was done at the sentence level, and the top three retrieval outputs for each query molecule are described. The corresponding description from the data pair was correctly retrieved at first for all cases, and the other retrieved candidates show a weak correlation with the query molecule.

## B.3 TEXT-TO-MOLECULE GENERATION RESULTS

Table 6 contains the performance of LDMol and other baselines for text-to-molecule generation on the PCdes test set, which is another prepared text-molecule paired dataset with detailed and specified descriptions. Note that LDMol was not trained with this test split of PCdes. Here, we only included some of the baselines in Table 1 which showed the highest performances with open access. Similar to the results on the ChebI-20 test split, LDMol outperformed the other transformer-based models on almost all metrics.

Figure 11 shows the behavior of LDMol's text-to-molecule generation with several exceptional scenarios. When we fed a completely ambiguous input such as *"beautiful"* or *"important"*, the model

Table 6: Benchmark results of text-to-molecule generation on PCdes test set. The best performance for each metric was written in **bold.**

| Model | Validity↑ | BLEU↑ | Levenshtein↓ | MACCS FTS↑ | RDK FTS↑ | Morgan FTS↑ | Match↑ | FCD↓ |
|---|---|---|---|---|---|---|---|---|
| MolT5$_{large}$ | 0.944 | 0.692 | 18.481 | 0.810 | 0.741 | 0.699 | 0.440 | 0.70 |
| bioT5 | **1.000** | 0.754 | 15.658 | 0.797 | 0.726 | 0.677 | 0.455 | 0.69 |
| bioT5+ | 0.999 | 0.677 | 20.464 | 0.743 | 0.615 | 0.541 | 0.266 | 1.09 |
| LDMol | 0.944 | **0.857** | **8.726** | **0.885** | **0.817** | **0.780** | **0.464** | **0.32** |

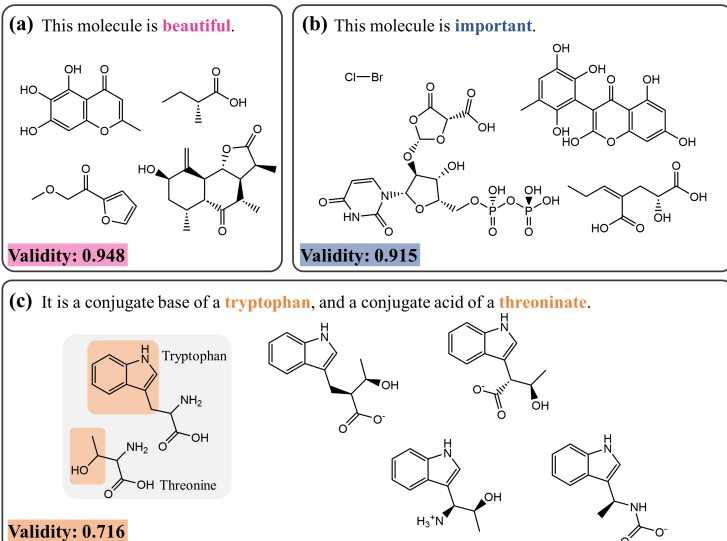

Figure 11: Examples of the generated molecules by LDMol, with (a, b) ambiguous text conditions and (c) contradictory and unreasonable input.

spits out a variety of different molecules without any consistency. When we fed contradictory inputs that could not be satisfied, the outputs were chimeric between contradictory prompts with a clearly decreased validity.

