# OpenReview forum: "LDMol: Text-to-Molecule Diffusion Model with Structurally Informative Latent Space"
_ICLR.cc/2025/Conference — Submitted to ICLR 2025_

### Official Review · Reviewer_UnHu · 2024-11-01

**Soundness:** 2
**Presentation:** 2
**Contribution:** 2
**Rating:** 5
**Confidence:** 2

**Summary:**

The paper presents LDMol, a latent diffusion model for text-to-molecule generation. LDMol leverages a structurally informative latent space to enhance the generation of molecules conditioned on natural language. The model incorporates a molecule autoencoder and utilizes contrastive learning to improve its feature extraction capabilities. The results suggest that LDMol outperforms existing autoregressive models and demonstrates versatility in tasks like molecule-to-text retrieval and text-guided molecule editing.

**Strengths:**

1. The use of a latent diffusion model for text-conditioned molecule generation is novel and addresses the discreteness of molecular data.
2. The application of contrastive learning to capture structural features of molecules is well-motivated and effectively implemented.
3. LDMol shows superior performance over baseline models in generating valid SMILES that align well with text conditions.

**Weaknesses:**

1. The model architecture, including the use of multiple components like autoencoders and diffusion models, might be complex to reproduce and understand without extensive background knowledge.

2. While the model performs well on specific datasets, its generalization to other types of chemical data or broader text inputs could be further examined.

3. Although the paper compares LDMol to existing models, more detailed comparisons with a wider range of benchmarks could strengthen the claims.

4. The reliance on large datasets for training might limit the applicability of the model in scenarios where such data is unavailable.

**Questions:**

1. How does the model handle ambiguous or contradictory natural language inputs?

2. Can the approach be adapted for other domains beyond molecules, such as protein or material generation?

3. What are the computational requirements for training and deploying LDMol compared to traditional autoregressive models?

4. Are there any limitations in terms of the types of molecules that can be generated using this model?

---

> ### Author Response · Authors · 2024-11-20
>
> We sincerely thank the reviewer for the constructive and valuable comments, and please refer to our responses below.
>
> **W1: Complex model pipeline of LDMol.**
>
> We would like to assure the reviewer that although LDMol comprises several modules, the relationship and information flow between them are very simple and intuitive. And since each module(AE, diffusion model, condition encoder) works independently, one can easily replace or improve each module. Given that StableDiffusion(which has an almost identical workflow) is also widely used and fine-tuned by many, we believe that LDMol is highly accessible to end users.
>
> **W2: Generalization ability on different types of data.**
>
> We would like to assure the reviewer that our model is not overfitted to any particular dataset. Rather it showed high performance in generation(Figure 4) and retrieval(Table 2) with short and less specific inputs, even in the relative out-of-domain distribution. Also, please refer to our text-to-molecule generation comparison in other test datasets in the next response.
>
> **W3: Request for a wider range of benchmarks.**
>
> Thank you for the suggestion of an additional benchmark. However, since text-to-molecule(T2M) generation is a recently emerging field, there is a lack of appropriate benchmarks other than CHEBI-20, and almost all the papers presented so far conducted their benchmark study on this. That said, we evaluated LDMol and several baselines' T2M generation performances on the PCdes test set, which is another prepared dataset with detailed captions. Note that LDMol was not trained with these data. We found that our model still outperforms the other baselines, and added this result to Appendix B.3(Table 6) in the revised manuscript.
>
> |Model | Validity | BLEU | Levenshtein | MACCS FTS | RDK FTS | Morgan FTS | Match | FCD |
> |:--|:-:|:-:|:-:|:-:|:-:|:-:|:-:|:-:|
> | MolT5$_{large}$ | 0.944 | 0.692 | 18.481 | 0.810 | 0.741 | 0.699 | 0.440 | 0.70 |
> | bioT5 | **1.000** | 0.754 | 15.658 | 0.797 | 0.726 | 0.677 | 0.455 | 0.69 |
> | bioT5+ | 0.999 | 0.677 | 20.464 | 0.743 | 0.615 | 0.541 | 0.266 | 1.09 |
> | LDMol  | 0.944 | **0.857** | **8.726** | **0.885** | **0.817** | **0.780** | **0.464** | **0.32** |
>
> **W4: The model's possible reliance on large datasets.**
>
> In fact, the 320k molecule-text paired data that LDMol was trained with is relatively very small compared to the other baselines; molXPT[1] and biot5+[2] used 8M and 3.2M molecule-text combined data from web, on top of other millions of natural language data. On the other hand, we only utilized a relatively small but well-curated dataset and still achieved high performance.
>
> **Q1: The model behavior on ambiguous or contradictory inputs.**
>
> When we fed a completely ambiguous input, the model spits out a variety of different but incoherent molecules with high validity. When we fed completely contradictory inputs that the model cannot follow, the output validity dropped significantly and the outputs were chimeric between contradictory prompts, which we regard as appropriate behavior. Please refer to the provided examples in Appendix B.3(Figure 11) in the revised manuscript.
>
> **Q2: Possibility for other important chemical domains.**
>
> Thanks for the insightful question, and we believe that our approach to the latent diffusion model is applicable to other important chemical data domains like materials and proteins. Here, note that our main contribution is to show that the latent space should not just be learnable, but also informative about the data domain. Therefore, a special encoder objective function tailored to the target domain could be suggested, just like we trained our encoder with contrastive learning to introduce structure invariants.
>
> **Q3: Computational requirements for training / deploying LDMol.**
>
> In the training phase, our model did not use more than 24GB of VRAM, so it can be trained on NVIDIA Geforce 3090. Meanwhile, molXPT and bioT5+ were trained on NVIDIA A100 with 80GB of VRAM (probably due to their large amount of training data). For the inference stage, our model can be used with 5GB of VRAM, which is not so different from our baselines of 2GB(bioT5+) and 4GB(molT5).
>
> **Q4: Possible limitations on the generated molecules.**
>
> Since our autoencoder is only trained on SMILES below a certain token length, LDMol does not generate output for several hundreds of tokens such as polymers. However, since diffusion models have already shown great scalability in the image domain, we think that the token length can be increased and this will not be a fundamental limitation in the future.
>
> ---
>
> [1] MolXPT: Wrapping Molecules with Text for Generative Pre-training, ACL 2023, arxiv:2305.10688
>
> [2] BioT5+: Towards Generalized Biological Understanding with IUPAC Integration and Multi-task Tuning, ACL 2024, arxiv:2402.17810

---

> ### Author Response · Authors · 2024-11-24
> **[Reminder] Summarization of our rebuttal**
>
> Dear reviewer UnHu,
>
> We’d like to summarize our revisions that addressed the questions and concerns you have raised. Specifically,
>
> 1. We ensured the proposed LDMol can be **trained with a much smaller dataset** and only needs **comparable computational requirements** compared to recent baselines.
>
> 2. We conducted extensive additional experiments for **output behavior for ambiguous and contradictory inputs** and **outperforming baselines in another benchmark** for LDMol's generalization ability.
>
> 3. We emphasized that LDMol is **not so complex compared to other widely-used latent diffusion models**, and has **great potential in many important chemical fields as one of the first diffusion models outperforming AR models in chemical instance generation**.
>
> As the deadline for the reviewer-author discussion phase is less than 72 hours away, we’d like to gently remind you and ask if we’ve adequately addressed your questions and concerns. We’re happy to provide clarification on any remaining questions you may have.
>
>
> Best regards,
> Authors

---

> ### Author Response · Authors · 2024-12-01
> **A kind reminder for discussion period deadline**
>
> Dear Reviewer UnHu,
>
> We are writing to kindly remind you regarding our responses to your comments on our manuscript. We comprehensively addressed your comments and questions with rich additional experiments, and emphasized our contribution with its broad applicability to other domains.
>
> As the deadline for finalizing the reviews is approaching within 48 hours, we would greatly appreciate it if you could timely consider our revision and inform us of your decisions. Please let us know if there are any further clarifications needed.
>
> Best regards, Authors

---

### Official Review · Reviewer_CrH4 · 2024-11-03

**Soundness:** 3
**Presentation:** 3
**Contribution:** 3
**Rating:** 5
**Confidence:** 3

**Summary:**

This study introduces LDMol, a latent diffusion model specifically designed for text-conditioned molecule generation. LDMol integrates a chemically informed autoencoder and employs contrastive learning to maintain structural consistency within the latent space, addressing issues with multiple SMILES representations for identical molecules. Experimental results indicate that LDMol surpasses current models in SMILES validity, structural fidelity, and condition adherence. Furthermore, LDMol proves versatile for applications such as molecule-to-text retrieval and text-guided molecule editing, offering a structurally aware diffusion model as a robust alternative to autoregressive methods in molecular generation research.

**Strengths:**

1.	It offers a thorough background on the difficulties of integrating text-based conditions with the discrete nature of molecular data, clearly outlining how LDMol tackles these challenges.
2.	The language and presentation are clear and scientifically rigorous, aligning well with standards for peer-reviewed publication.
3.	The authors substantiate their claims of enhanced text-conditioned generation through extensive benchmarking and showcase LDMol's applicability in downstream tasks, reinforcing its potential impact.

**Weaknesses:**

1. While the contrastive learning approach with SMILES enumeration is well-justified and innovative, a comparative discussion on why prior methods struggled with achieving structural consistency would enhance the reader's understanding of LDMol’s novelty.

2. The manuscript could enhance clarity by detailing the hyperparameter tuning process for the contrastive loss, especially concerning the InfoNCE loss temperature τ and its impact on the latent space.

3. The manuscript uses a linear compression layer to reduce the dimensionality of the latent space, which is sensible to avoid overfitting. The authors should, however, provide a more thorough justification of the chosen dimension (64) and discuss alternatives.

4. To refine the SMILES encoder’s capacity to distinguish between subtle structural variations (e.g., stereoisomers), it would be beneficial to conduct experiments incorporating additional molecular descriptors (e.g., 3D structural descriptors). This may improve encoding accuracy for complex molecular structures.

5. Some related works are needed to discuss in the manuscript, such as [1-2].

[1]Grisoni F, Moret M, Lingwood R, et al. Bidirectional molecule generation with recurrent neural networks[J]. Journal of chemical information and modeling, 2020, 60(3): 1175-1183.

[2] Liu X, Guo Y, Li H, et al. DrugLLM: Open Large Language Model for Few-shot Molecule Generation[J]. arXiv preprint arXiv:2405.06690, 2024.

**Questions:**

See above.

---

> ### Author Response · Authors · 2024-11-20
>
> We sincerely thank the reviewer for the insightful and constructive comments. Please refer to our responses below.
>
> **W1: Discussion about achieving structural consistency in the latent space.**
>
> Thanks for the important comments that highlight our innovation. We found that naive VAE shows structural inconsistency because knowing how to reconstruct input doesn't mean a complete understanding of the data domain. Even in LLMs with Autoregressive modeling, it is known that even if the model has successfully learned the grammar of the natural language, the model doesn't understand its meaning and often hallucinates nonsensical output[1]. Similarly, VAE is prone to only reconstruct the SMILES strings while not knowing what atoms and bonds the input SMILES imply. We added this discussion in the revised manuscript(line 239~243).
>
> **W2: Hyperparameter for the encoder contrastive learning.**
>
> Thanks for the constructive comments. It is known that lower temperature (tau) in contrastive learning penalizes hard negatives more strongly, making the learned feature more sensitive to fine-grain details[2]. We considered this as an appropriate property for our latent space and used a small tau of 0.07. Using too big tau reduced the model's ability to distinguish different molecules and made the training loss converge to a much higher value, but the values around 0.07 work well. This result can be found in Appendix A.2(Figure 7) in the revised manuscript.
>
> **W3: Justification and possible alternatives for the linear compression layer.**
>
> Thanks for giving an opportunity to clarify. We determined the feature dimension empirically by monitoring reconstruction accuracy. Please refer to the table below, which we included in Appendix A.2(Table 5) in the revised manuscript.
>
> |Linear compression size $d_z$ | Recon. acc. |
> |:--|:-:|
> | $d_z$=32 | 0.948 |
> | $d_z$=64 | 0.980 |
> | $d_z$=128 | 0.989 |
>
> When we compressed it into 32, the reconstruction rate dropped to 94.8%. When we compressed it into 128, the reconstruction rate was slightly higher, but the training time for the subsequent diffusion model increased quadratically. Considering that the failed 2% for the current model were mostly very long molecules, we decided $d_z$ to 64.
>
> Meanwhile, an alternative compression method could be a transformer layer such as Perceiver-Resampler[3][4]. However, we had to keep our compression layer as simple as possible, because adding another complex layer makes the latent space deviate from the former informative and well-regulated learnable space. In fact, when we replaced our linear layer with the transformer module, the performance was significantly decreased as shown below, as the output space became less learnable.
>
> |Model | Validity | BLEU | Levenshtein | MACCS FTS | RDK FTS | Morgan FTS | Match | FCD |
> |:--|:-:|:-:|:-:|:-:|:-:|:-:|:-:|:-:|
> LDMol w/ transformer compression | 0.565 | 0.783 | 18.607 | 0.846 | 0.699 | 0.661 | 0.084 | 2.19 |
> LDMol  | **0.941** | **0.926** | **6.750** | **0.973** | **0.950** | **0.931** | **0.530** | **0.20** |
>
> And considering that our linear layer has much fewer parameters(66K) than the transformer module(22M), we concluded that the linear compression was sufficient for our work.
>
> **W4: Incorporating additional 3d descriptors for performance improvement.**
>
> Thank you for your valuable suggestion, and we acknowledge that we could improve our model performance by injecting more external information into the constructed latent space. Unfortunately, we have not yet found a trainable 3d descriptor database that provides structural information beyond what SMILES already provides. Also, there should be another contribution to combine that information with the current latent space as a structural invariant(e.g. features from coordinate value should be SE(3)-invariant, etc.), so we found that this is beyond the scope of our current manuscript. We emphasize our current work as the first successful text-to-molecule diffusion models outperforming AR baselines, and hope you understand that we leave this approach as future work.
>
> **W5: Discussion about several related works.**
>
> Thank you for the suggestion, and we have added the works for molecule generation using bidirectional RNNs [5] and LLMs [6] into the Related Works section of the manuscript(line 146~154).
>
> ---
>
> [1] A survey on hallucination in large language models: Principles, taxonomy, challenges, and open questions, arxiv:2311.05232
>
> [2] Understanding the Behaviour of Contrastive Loss, CVPR 2021, arxiv:2012.09740
>
> [3] Flamingo: a Visual Language Model for Few-Shot Learning, NeurIPS 2022, arxiv:2204.14198
>
> [4] BLIP-2: Bootstrapping Language-Image Pre-training with Frozen Image Encoders and Large Language Models, PMLR 2023, arxiv:2301.12597
>
> [5] Bidirectional molecule generation with recurrent neural networks. Journal of chemical information and modeling 2020
>
> [6] DrugLLM: Open Large Language Model for Few-shot Molecule Generation, arxiv:2405.06690.

---

> ### Author Response · Authors · 2024-11-24
> **[Reminder] Summarization of our rebuttal**
>
> Dear reviewer CrH4,
>
> We’d like to summarize our revisions that addressed the questions and concerns you have raised. Specifically,
>
> 1. We included **theoretical support and experimental demonstration of the temperature(tau) selection** for the encoder training.
>
> 2. We explained **the choice of our linear compression dimension size according to the reconstruction rate** and discussed possible alternatives with various experiments.
>
> 3. We followed your suggestions and **included more related works and extensive discussions about achieving structurally consistent latent space**.
>
> As the deadline for the reviewer-author discussion phase is approaching less than 72 hours, we would like to extend a warm reminder and ask whether we have addressed your questions and concerns adequately. We would be happy to clear up any additional questions.
>
> Best regards,
> Authors

---

> ### Author Response · Authors · 2024-12-01
> **A kind reminder for discussion period deadline**
>
> Dear Reviewer CrH4,
>
> We are writing to kindly remind you regarding our responses to your comments on our manuscript. We addressed your concerns on our model with thorough experimental results, and revised our manuscript with additional discussions to support and emphasize our contribution.
>
> As the deadline for finalizing the reviews is approaching within 48 hours, we would greatly appreciate it if you could timely consider our revision and inform us of your decisions. Please let us know if there are any further clarifications needed.
>
> Best regards, Authors

---

### Official Review · Reviewer_aYeM · 2024-11-04

**Soundness:** 3
**Presentation:** 3
**Contribution:** 2
**Rating:** 5
**Confidence:** 3

**Summary:**

* The authors propose LDMol, a latent diffusion model for text-to-molecule generation.
* A contrastive learning method is applied to learn a latent representation for molecules, and an LDM (with DiT backbone) is trained over the compressed latent space.
* The generative pipeline finally involves feeding the latent produced by the LDM to a separate AR decoder.

**Strengths:**

* The use of contrastive learning to learn SMILE representations that are invariant to traversal order is convincing and seems well-motivated.
* Solid evaluation of their proposed method featuring a lot of existing baselines on a variety of metrics.
* There is a clear need and desire in the community for more works on diffusion generation, especially on discrete data domains, rendering this work relevant and timely.

**Weaknesses:**

* It seems odd that an AR transformer is needed for decoding when one of the selling points of LDM / diffusion models is parallel generation that relaxes the AR factorization of probability. The need for an LDM as well as a separately trained AR decoder makes the pipeline more convoluted in comparison to the usual domains (e.g., LDM for text-to-image generation).
* The work seems to largely apply existing techniques proposed in the image / text-to-image literature with little to moderate modifications for adaptation to the molecule generation domain.

**Questions:**

* Why is a separate linear compressor layer necessary? Why not configure the size of the latent space for the SMILE encoder / decoders to be smaller to begin with?
* (Related to above) Why is it necessary to train a separate AR transformer for the decoding step? Is it possible to feed the DiT output back into the SMILE decoder to generate a molecule sequence? The linear compressor layer is not invertible so this is probably not possible in the current setup, but assuming it were, would it not be simpler to use the trained decoder?
* Relative to compared baselines, is the proposed method faster? How does the runtime of the backward diffusion step compare to the AR decoding step?

---

> ### Author Response · Authors · 2024-11-20
>
> We sincerely thank the reviewer for thorough reading and insightful comments, and please refer to our responses below.
>
> **W1: Convoluted pipeline with AR transformer.**
>
> We like to assure the reviewer that no additional complex models are added compared to the mentioned text-to-image(T2I) LDM. Both comprise an autoencoder(AE) and a conditional diffusion model, where the decoder part of our AE is an autoregressive(AR) transformer. This AR decoder doesn't model the relationship between text and molecules at all, and only recovers SMILES strings from given latent. We employed an AR transformer as our decoder because SMILES itself is sequential data with a varied length. Since our decoder is only used once at the end of sampling without a special AR sampling technique(*e.g.* beam search), we would like to assure the reviewer that the required time for LDMol was not so long compared to the other baselines.
>
> **W2: Modification on top of the existing models.**
>
> We agree that LDMol indeed has a similar pipeline to T2I latent diffusion models. However, our biggest contribution is preparing a latent space that embeds the useful and consistent chemical information, improving the diffusion model performance. This allows LDMol to be one of the first diffusion models outperforming AR models in sequence generation. Note that without our contrastive encoder training, the model trained on the naive latent space did not surpass the previous baselines(Table 3). We believe that the proposed SMILES-enumeration contrastive learning is a domain-specific and novel approach to encoding molecular information.
>
> **Q1: The necessity of the separate linear compression layer.**
>
> The reason we didn't create an encoder with the compressed dimension from the beginning is that we trained our encoder with contrastive learning to extract as informative and structure-invariant features as possible. When we directly applied contrastive loss with 64-dimensioned features, the trained encoder failed to learn informative features, and the corresponding decoder couldn't reconstruct the input molecule. Please refer to the table below, which we also included in Appendix A.2(Table 5) in the revised manuscript.
>
> |Methods | Reconstruction accuracy |
> |:--|:-:|
> |Contrastive learning directly on the compressed latent of $d_z$=64 | 0.084 |
> |Contrastive learning with $d_{enc}$=1024, then compress to $d_z$=64 | 0.980 |
>
> **Q2: The usage of a separate AR transformer besides of the trained SMILES decoder.**
>
> We want to emphasize that we only trained a single SMILES decoder, and didn't build a "separate" autoregressive transformer; the trained SMILES decoder is an autoregressive transformer. You can consider "SMILES encoder + linear compression layer" as the total encoder part of our AE, and "autoregressive transformer" as the decoder part. Therefore, the suggested “feeding the DiT output directly back into the SMILES decoder (via cross-attention)” is what we’re doing currently.
>
> **Q3: The speed and runtime of the proposed model.**
>
> For the inference time, despite the usage of the latent decoder and Classifier-Free Guidance (CFG) that doubles the diffusion model iteration, LDMol showed comparable inference speed to other AR transformer models; as shown in the table below, LDMol was slower than bioT5+ but faster than the model of molT5$_{large}$. Furthermore, various studies and techniques have been proposed to reduce the diffusion model inference speed [1][2], so we believe that future research can further address the issue of LDMol’s inference time.
>
> | | molT5$_{large}$ |bioT5+| LDMol(ours) |
> |:--|:-:|:-:|:-:|
> |required VRAM| < 4GB | < 2GB | < 5GB |
> |Inference time(1,000 samples, batch size=10)| 523s | 180s | 361s |
>
>  About the required time in the reverse diffusion and AR decoding step, we observed that about 70% of the runtime is used for the reverse diffusion and 30% for the AR decoding steps.
>
> ---
>
> [1] SDXL: Improving Latent Diffusion Models for High-Resolution Image Synthesis, ICLR 2024, arxiv:2307.01952
>
> [2] Consistency models, ICML 2023, arxiv:2303.01469

---

> ### Author Response · Authors · 2024-11-24
> **[Reminder] Summarization of our rebuttal**
>
> Dear reviewer aYeM,
>
> We’d like to summarize our revisions that addressed questions and concerns you have raised. Specifically,
>
> 1. We emphasized our biggest contribution which is **constructing a novel latent space to embed the useful and consistent chemical information**, and demonstrating **this can improve the performance of the diffusion model** on it.
>
> 2. We rationalized **our AR decoder as a replacement of the conventional latent decoders for length-varied SMILES**, and **justified the introduction of the linear compression layer** with an experiment.
>
> 3. We showed that **LDMol can operate with a comparable inference speed with current baseline models**, even with the usage of the latent decoder and CFG.
>
> As the deadline for the reviewer-author discussion phase is less than 72 hours away, we’d like to gently remind you and ask if we’ve adequately addressed your questions and concerns. We’re happy to provide clarification on any remaining questions you may have.
>
> Best regards,
> Authors

---

> ### Author Response · Authors · 2024-12-01
> **A kind reminder for discussion period deadline**
>
> Dear Reviewer aYeM,
>
> We are writing to kindly remind you regarding our responses to your comments on our manuscript. We have assured to resolve your concerns about our model design choices, and emphasized our contribution to novel molecule latent space enabled a successful text-to-molecule diffusion model.
>
> As the deadline for finalizing the reviews is approaching within 48 hours, we would greatly appreciate it if you could timely consider our revision and inform us of your decisions. Please let us know if there are any further clarifications needed.
>
> Best regards, Authors

---

### Official Review · Reviewer_A8d4 · 2024-11-04

**Soundness:** 3
**Presentation:** 3
**Contribution:** 3
**Rating:** 6
**Confidence:** 3

**Summary:**

The paper presents LDMol, a latent diffusion model for generating molecules based on text conditions. It overcomes the challenges in aligning molecular discreteness with natural language descriptions. LDMol employs a SMILES autoencoder that uses contrastive learning to create a chemically informative and structurally consistent latent space.

**Strengths:**

1. By incorporating contrastive learning into the SMILES autoencoder, LDMol achieves a chemically informative latent representation.
2. LDMoI proves effective in related applications like molecule-to-text retrieval and text-guided molecule editing, demonstrating the model's adaptability and potential utility across various domains within chemical and biomedical research.

**Weaknesses:**

1. The performance relies on text-molecule paired datasets, which are often limited to structural descriptors.
2. LDMol relies on detailed text prompts to generate precise molecular structures, which may reduce its effectiveness with vague or less specific text conditions.

**Questions:**

1. Since LDMol relies on paired text-molecule data, how does the model perform with smaller or less detailed datasets? Could the model be trained effectively on a smaller dataset, and are there any plans to explore data efficiency?
2. The model struggles with distinguishing stereoisomers effectively. What impact does this limitation have on downstream tasks like molecule editing or retrieval? Are there specific modifications that could improve the model’s stereoisomer sensitivity?
3. How does the computational efficiency of LDMol compare to SOTA autoregressive models?

---

> ### Author Response · Authors · 2024-11-20
>
> We sincerely thank the reviewer for thorough reading and insightful comments. Please refer to our replies to your concerns below.
>
> **W1: Text-molecule paired datasets are often limited to structural descriptors.**
>
> We agree that the prepared dataset had fewer descriptions for bonds and atoms. However, even in text-molecule paired data set there were still a number of descriptions for functional groups and motifs which enabled LDMol to achieve SOTA-level generation performance. Given that text-conditioned molecule generation is a recently emerging field and high-quality large-scale text-molecule pair data is only now being presented, we believe our diffusion model methodology of LDMol can be further advanced in future works by incorporating more structurally and biochemically enriched data.
>
> **W2: Effectiveness of LDMol with vague or less specific text conditions.**
>
> We would like to assure the Reviewer that many text descriptions in the PubchemSTM training dataset were very short and vague; 29% of the PubchemSTM captions have less than 50 characters, and 68% of the captions have less than 100 characters. We emphasize that LDMol could generate molecules with both highly detailed conditions (Table 1) and less specific prompts (Figure 4). LDMol also performed well in the retrieval downstream task which includes unseen and out-of-domain text, indicating that our model's comprehension ability was not overfitted to the training data.
>
> **Q1: Model performance & training efficiency for small and less-detailed datasets.**
>
> Most recent transformer-based baselines used millions of single-modal and multimodal data from various databases. For example, Biot5+[1] used 8M and 3.2M molecule-text combined data from web crawling on top of millions of natural language data. MolT5[2] used 100 million molecule data to model the SMILES data distribution. Compared to these works, the 320k text-molecule pairs we introduced are actually a very small amount of data, and yet LDMol outperformed all baselines.
>
> **Q2: Modifications for the model's stereoisomer sensitivity.**
>
> We first emphasize that our introduced hard-negative samples in the encoder training were sufficiently helpful for LDMol to distinguish subtle structure differences such as stereoisomers. More specifically, we found that our LDMol encoder distinguishes the given molecule's stereoisomer from the SMILES enumeration of itself in 87.7% of the cases, whereas 50.3% for the encoder trained without hard-negatives. This improved the exact match ratio for the text-to-molecule generation benchmark by more than 25%p(Table 3), enabling LDMol to outperform previously suggested models.
>
> That said, the stereoisomer discrimination problem is definitely a hindrance to the model performance, when downstream tasks require very subtle structure differences. We expect that injecting explicit 3-dimensional descriptor information (e.g. coordinates) effectively into the latent space might further resolve this problem, which could be an interesting future work.
>
> **Q3: Computational efficiency of LDMol.**
>
> Please refer to the table below that compares the computational efficiency of LDMol and several baselines.
>
> | | molT5$_{large}$ |bioT5+| LDMol(ours) |
> |:--|:-:|:-:|:-:|
> |required VRAM| < 4GB | < 2GB | < 5GB |
> |Inference time(1,000 samples, batch size=10)| 523s | 180s | 361s |
>
> In terms of memory usage, our model can operate with less than 5GB of VRAM, which is comparable to other transformer models with 2GB for biot5+ and 4GB of molT5_large. The required time was also comparable to transformer-based models, even with the latent decoder and the Classifier-Free Guidance(CFG) that doubles the diffusion model usage. Also, many works have been published to reduce the inference time of diffusion models [3][4]. Therefore, as one of the first successful text-to-molecule diffusion models, we believe that the inference time can be further improved by future research.
>
> ---
> [1] BioT5+: Towards Generalized Biological Understanding with IUPAC Integration and Multi-task Tuning, ACL 2024, arxiv:2402.17810
>
> [2] Translation between molecules and natural language, EMNLP 2022, arxiv:2204.11817
>
> [3] Accelerating parallel sampling of diffusion models, ICML 2024, arxiv:2402.09970
>
> [4] Consistency models, ICML 2023, arxiv:2303.01469

---

> ### Author Response · Authors · 2024-11-24
> **[Reminder] Summarization of our rebuttal**
>
> Dear reviewer A8d4,
>
> We’d like to summarize the addressed questions and concerns you have raised. Specifically,
>
> 1. We ensured that **LDMol is trained with a much smaller amount of data including vague texts** compared to previous works, and yet **able to generate molecules that faithfully follow the given condition**, from less-detailed texts to highly specific prompts.
>
> 2. We explained how **our introduced stereoisomer hard-negatives improved the sensitivity of the constructed latent space** and suggested possible improvements in further research.
>
> 3. We showed that *LDMol can operate with efficient time and memory usage** which is comparable to baseline models.
>
> As the deadline for the reviewer-author discussion phase is approaching less than 72 hours, we would like to extend a warm reminder and ask whether we have addressed your questions and concerns adequately. We would be happy to clear up any additional questions.
>
> Best regards,
> Authors

---

> > ### Comment · Reviewer_A8d4 · 2024-11-26
> >
> > Thanks for your clearification. I will keep the score.

---

> > > ### Author Response · Authors · 2024-11-26
> > >
> > > Thank you for your thoughtful consideration of our paper, and we are delighted that you scored our paper as acceptance of our work.
> > >
> > > We are also committed to addressing any further comments or suggestions you may have to enhance the quality of our manuscript.

---

### Author Response · Authors · 2024-11-20
**General Response**

We would like to thank all the reviewers for their constructive and thorough reviews.
We are encouraged that the reviewers view our paper as **achieving chemically informative latent representation**(A8d4), **meets the successful diffusion generation on discrete data domains**(aYeM), **clear and scientifically rigorous presentation**(CrH4), and **showing superior text-to-molecule generation and downstream performance over baseline models**(UnHu).

We have made the following major revisions to address the concerns of the reviewers:

**1. Analysis of the used dataset size and computational efficiency.**

To resolve the reviewers'(A8d4, aYeM, UnHu) concerns, we showed that our model achieved great text-to-molecule generation performance with a relatively small amount of data, and can be trained and used with similar time and resources with transformer-based baselines.

**2. More discussion about the limitation of naively constructed latent space.**

We extended our discussion about the previous works and the structural inconsistency of the latent space from naive VAEs, which highlights the novelty of our works on constructing chemically informative latent.

**3. Justification of our model design choices.**

In response to the reviewers'(aYeM, CrH4) suggestions, we conducted various experiments to support our model design selection for the temperature parameter tau, the introduction of the linear compression layer, its feature size and possible alternatives.

**4. Further experiments on additional benchmark datasets and more variate text prompts.**

We now include more text-to-molecule generation benchmark results on PCdes[1] to demonstrate that LDMol still outperforms the baselines, and the molecule generation examples with more wide range of text inputs.

For point-to-point responses, please refer to below.

---

[1] A deep-learning system bridging molecule structure and biomedical text with comprehension comparable to human professionals, Nature Communications 13, Article number: 862 (2022)

---

### Author Response · Authors · 2024-11-27
**Kind reminder for the manuscript PDF revision deadline**

Dear reviewers,

We have addressed the questions from the reviewers in the comments with various additional results, and revised the paper accordingly.

Since the deadline for the manuscript revision is closing very soon (less than 24 hours), we'd like to kindly remind the reviewers to consider our responses and provide further suggestions if they require the revision of the manuscript.

Best regards, Authors

---

### Meta-Review · Area_Chair_Rxqs · 2024-12-19

**Metareview:**

This paper proposes a latent diffusion model for text-conditioned molecule generation that uses contrastive learning to create a chemically informative latent space. The key strengths are the novel application of diffusion models to molecular generation, effective use of contrastive learning for structural consistency, and strong empirical results surpassing autoregressive baselines on various metrics. Several critical weaknesses emerge from the reviews: (1) The model architecture appears overly complex, requiring both a diffusion model and an autoregressive decoder, which seems to negate the parallel generation benefits of diffusion models, (2) The approach largely applies existing techniques from text-to-image generation with only moderate adaptations to molecules, (3) The evaluation lacks comprehensive benchmarking across diverse molecular datasets and conditions, and (4) There are concerns about the model's reliance on detailed text prompts and handling of stereoisomers. Given these limitations and the limited the technical contributions, the paper may fall short of ICLR's bar for acceptance.

**Additional Comments On Reviewer Discussion:**

The authors provided extensive experimental results justifying their architectural choices, including ablations on the linear compression layer and comparisons of inference speed with baselines. They also added evaluations on an additional benchmark dataset and analyzed the model's behavior with ambiguous inputs. However, only one reviewer (A8d4) was convinced enough to maintain a positive score, while the other three kept their marginal reject recommendations. The authors' responses, while thorough, did not fully address the fundamental concern about the novelty of applying existing diffusion techniques to molecules. I have encouraged the reviewers for a discussion, but it seems the reviewers did not respond to the authors' final clarifications, thus I cannot assume their concerns have been resolved. Weighing the limited technical novelty against the solid empirical results, I maintain the rejection recommendation, but my decision can be bumped up.

---

### Decision · Program_Chairs · 2025-01-22

Reject